# Alternating Optimization of Decision Trees, with Application to Learning Sparse Oblique Trees

**Miguel Á. Carreira-Perpiñán**
Dept. EECS, University of California, Merced
`mcarreira-perpinan@ucmerced.edu`

**Pooya Tavallali**
Dept. EECS, University of California, Merced
`ptavallali@ucmerced.edu`

## Abstract

Learning a decision tree from data is a difficult optimization problem. The most widespread algorithm in practice, dating to the 1980s, is based on a greedy growth of the tree structure by recursively splitting nodes, and possibly pruning back the final tree. The parameters (decision function) of an internal node are approximately estimated by minimizing an impurity measure. We give an algorithm that, given an input tree (its structure and the parameter values at its nodes), produces a new tree with the same or smaller structure but new parameter values that provably lower or leave unchanged the misclassification error. This can be applied to both axis-aligned and oblique trees and our experiments show it consistently outperforms various other algorithms while being highly scalable to large datasets and trees. Further, the same algorithm can handle a sparsity penalty, so it can learn *sparse oblique trees*, having a structure that is a subset of the original tree and few nonzero parameters. This combines the best of axis-aligned and oblique trees: flexibility to model correlated data, low generalization error, fast inference and interpretable nodes that involve only a few features in their decision.

## 1 Introduction

Decision trees are among the most widely used statistical models in practice. They are routinely at the top of the list in the `KDnuggets.com` annual polls of top machine learning algorithms and other top-10 lists [36]. Many statistical or mathematical packages such as SAS or Matlab implement them. Apart from being able to model nonlinear data well in the first place, they enjoy several unique advantages. The prediction made by the tree is a path from the root to a leaf consisting of a sequence of decisions, each involving a question of the type "$x_j > b_i$" (is feature $j$ bigger than threshold $b_i$?) for axis-aligned trees, or "$\mathbf{w}_i^T \mathbf{x} > b_i$" for oblique trees. This makes inference very fast, and may not even need to use all input features to make a prediction (with axis-aligned trees). The path can be understood as a sequence of IF-THEN rules, which is intuitive to humans, and indeed one can equivalently turn the tree into a database of rules. This can make decision trees preferable over more accurate models such as neural nets in some applications, such as medical diagnosis or legal analysis.

However, trees pose one crucial problem that is only partly solved to date: learning the tree from data is a very difficult optimization problem, involving a search over a complex, large set of tree structures, and over the parameters at each node. For simplicity, in this paper we focus on classification trees with a binary tree (having a binary split at each node) where the bipartition in each node is either an axis-aligned hyperplane or an arbitrary hyperplane (oblique trees). However, many of our arguments carry over beyond this case.

Ideally, the objective function we would like to optimize has the usual form of a regularized loss:
$$E(T) = \mathcal{L}(T) + \alpha\, C(T) \qquad \alpha > 0 \tag{1}$$
where $\mathcal{L}$ is the misclassification error on the training set, given below in eq. (2), and $C$ is the complexity of the tree, e.g. its depth or number of nodes. This is necessary to avoid large trees that finely

split the space so the dataset is perfectly classified, but would likely overfit. Finding an optimal decision tree is NP-hard [22] even if we fix its number of splits [26].

How does one learn a tree in practice (also called "tree induction")? After many decades of research, the algorithms that have stood the test of time are, in spite of their obvious suboptimality, (variations of) greedy growing and pruning, such as CART [8] or C4.5 [31]. First, a tree is grown by recursively splitting each node into two children, using an impurity measure. One can stop growing and return the tree when the impurity of each leaf falls below a set threshold. Even better trees are produced by growing a large tree and pruning it back one node at a time. At each growing step, the parameters at the node are learned by minimizing an impurity measure such as the Gini index, cross-entropy or misclassification error. The goal is to find a bipartition where each class is as pure (single-class) as possible. Gini or cross-entropy are preferable to misclassification error because the former are more sensitive to changes in the node probabilities than the misclassification rate [20, p. 309]. Minimizing the impurity over the parameters at the node depends on the node type:

- Axis-aligned trees: the exact solution can be found by enumeration over all (feature, threshold) combinations. For a given feature (= axis), the possible thresholds are the midpoints between consecutive training point values along that axis. For a node $i$ containing $N_i$ training points in $\mathbb{R}^D$, an efficient implementation can do this in $\mathcal{O}(DN_i)$ time.

- Oblique trees: minimizing the impurity is much harder because it is a non-differentiable function of the weights. As these change continuously, points change side of the hyperplane discontinuously, and so does the impurity. The standard approach is coordinate descent over the weights at the node, but this tends to get stuck in poor local optima [8, 28].

The optimization over the node parameters (exact for axis-aligned trees, approximate for oblique trees) assumes the rest of the tree (structure and parameters) is fixed. The greedy nature of the algorithm means that once a node is optimized, it its fixed forever.

Note that it is only in the leaves where an actual predictive model is fit. The internal nodes do not do prediction, they partition the space ever more finely into boxes (axis-aligned trees) or polyhedra (oblique trees). Each internal node bipartitions its region. Each leaf fits a local model to its region (for classification, the model is often the majority label of the training points in its region). Hence, the tree is really a partition of the space into disjoint regions with a local predictor at each region and a fast access to the region for a given input point (by propagating it through the tree).

The majority of trees used in practice are axis-aligned, not oblique. Two possible reasons for this are 1) an oblique tree is slower at inference and less interpretable because each node involves all features. And 2) as noted above and confirmed in our experiments, coordinate descent for oblique trees does not work as well, and often an axis-aligned tree will outperform in test error an oblique tree of similar size. This is disappointing because an axis-aligned tree imposes an arbitrary region geometry that is unsuitable for many problems and results in larger trees than would be needed.

In this paper we improve both of these problems with oblique trees. We focus on a restricted setting: we assume a given tree structure, given by an initial tree (CART or random). We propose an optimization algorithm for the tree parameters that considerably decreases its misclassification error. Further, this allows us to introduce a new type of trees that we call *sparse oblique trees*, where each node is a hyperplane involving only a small subset of features, and whose structure is a pruned version of the original tree. Our algorithm is based on directly optimizing the quantity of interest, the misclassification error, using alternating optimization over separable subsets of nodes. After a section 2 on related work, we describe our algorithm in section 3 and evaluate it in sections 4 and 5.

## 2  Related work

The CART book [8] is a good summary of work on decision trees up to the 80s, including the basic algorithms to learn the tree structure (greedy growing and pruning), and to optimize the impurity measure at each node (by enumeration for the axis-aligned case and coordinate descent over the weights for the oblique case). OC1 [28] is a minor variation of the coordinate descent algorithm of CART for oblique trees that uses multiple restarts and random perturbations after convergence to try to find a better local optimum, but its practical improvement is marginal. See [31, 32] for reviews of more recent work, including tree induction and applications. There is also a large literature on

constructing ensembles of trees, such as random forests [7, 13] or boosting [33], but we focus here on learning a single tree.

Much research has focused on optimizing the parameters of a tree given a initial tree (obtained with greedy growing and pruning) whose structure is kept fixed. Bennett [2, 3] casts the problem of optimizing a fixed tree as a linear programming problem, in which the global optimum could be found. However, the linear program is so large that the procedure is only practical for very small trees (4 internal nodes in her experiments); also, it applies only to binary classification. Norouzi et al. [29, 30] introduce a framework based on optimizing an upper bound over the tree loss using stochastic gradient descent (initialized from an already induced tree). Their method is scalable to large datasets, however it is not guaranteed to decrease the real loss function of a decision tree and may even marginally worsen an already induced tree.

Bertsimas and Dunn [4] formulate the optimization over tree structures (limited to a given depth) and node parameters as a mixed-integer optimization (MIO) by introducing auxiliary binary variables that encode the tree structure. Then, one can apply state-of-the-art MIO solvers (based on branch-and-bound) that are guaranteed to find the globally optimum tree (unlike the classical, greedy approach). However, this has a worst-case exponential cost and again is not practical unless the tree is very small (depth 2 to 4 in their paper).

Methods such as T2 [1], T3 [34] and T3C [35], introduce a family of efficient enumeration approaches constructing optimal non-binary decision trees of depths up to 3. These trees may not be as interpretable as binary ones and do not outperform existing approaches of inducing trees [34, 35].

Finally, soft decision trees assign a probability to every root-leaf path of a fixed tree structure, such as the hierarchical mixtures of experts [23]. The parameters can be learned by maximum likelihood with an EM or gradient-based algorithm. However, this loses the fast inference and interpretability advantages of regular decision trees, since now an instance must follow each root-leaf path.

## 3 Alternating optimization over node sets

**Problem definition**   We want to optimize eq. (1) but assuming a given, fixed tree structure (obtained e.g. from the CART algorithm, i.e., greedy growing and pruning for axis-aligned or oblique trees with impurity minimization at each node). Equivalently, since the tree complexity is fixed, we minimize the misclassification cost jointly over the parameters $\boldsymbol{\Theta} = \{\boldsymbol{\theta}_i\}$ of all nodes $i$ of the tree:

$$\mathcal{L}(\boldsymbol{\Theta}) = \sum_{n=1}^{N} L(y_n, T(\mathbf{x}_n; \boldsymbol{\Theta})) \tag{2}$$

where $\{(\mathbf{x}_n, y_n)\}_{n=1}^{N} \subset \mathbb{R}^D \times \{1, \ldots, K\}$ is a training set of $D$-dimensional real-valued instances and their labels (in $K$ classes), $L(\cdot, \cdot)$ is the 0/1 loss, and $T(\mathbf{x}; \boldsymbol{\Theta})$: $\mathbb{R}^D \rightarrow \{1, \ldots, K\}$ is the predictive function of the tree. This is obtained by propagating $\mathbf{x}$ along a path from the root down to a leaf, computing a binary decision $f_i(\mathbf{x}; \boldsymbol{\theta}_i)$: $\mathbb{R}^D \rightarrow \{\text{left}, \text{right}\}$ at each internal node $i$ along the path, and outputting the leaf's label. Hence, the parameters $\boldsymbol{\theta}_i$ at a node $i$ are:

- If $i$ is a leaf, $\boldsymbol{\theta}_i = \{y_i\} \subset \{1, \ldots, K\}$ contains the label at that leaf.
- If $i$ is an internal (decision) node, $\boldsymbol{\theta}_i = \{\mathbf{w}_i, b_i\}$ where $\mathbf{w}_i \in \mathbb{R}^D$ is the weight vector and $b_i \in \mathbb{R}$ the threshold (or bias) for the decision hyperplane "$\mathbf{w}_i^T \mathbf{x} \geq b_i$". For axis-aligned trees, the decision hyperplane is "$x_{k(i)} \geq b_i$" where $k(i) \in \{1, \ldots, D\}$, i.e., we threshold the feature $k(i)$, hence $\boldsymbol{\theta}_i = \{k(i), b_i\}$.

**Separability condition**   The key to design a good optimization algorithm for (2) is the following separability condition. Assume the parameters are not shared across nodes ($i \neq j \Rightarrow \boldsymbol{\theta}_i \cap \boldsymbol{\theta}_j = \varnothing$).
**Theorem 3.1** (separability condition). *Let $T(\mathbf{x}; \boldsymbol{\Theta})$ be the predictive function of a rooted directed binary decision tree. If nodes $i$ and $j$ (internal or leaves) are not descendants of each other, then, as a function of $\boldsymbol{\theta}_i$ and $\boldsymbol{\theta}_j$ (i.e., fixing all other parameters $\boldsymbol{\Theta} \setminus \{\boldsymbol{\theta}_i, \boldsymbol{\theta}_j\}$), the function $\mathcal{L}(\boldsymbol{\Theta})$ of eq. (2) can be written as $\mathcal{L}(\boldsymbol{\theta}_i, \boldsymbol{\theta}_j) = \mathcal{L}_i(\boldsymbol{\theta}_i) + \mathcal{L}_j(\boldsymbol{\theta}_j) + constant$, where the "constant" does not depend on $\boldsymbol{\theta}_i$ or $\boldsymbol{\theta}_j$.*

*Proof.* Each training point $\mathbf{x}_n$ for $n \in \{1, \ldots, N\}$ follows a unique path from the root to one leaf of the tree. Hence, a node $i$ receives a subset $\{(\mathbf{x}_n, y_n)\colon n \in \mathcal{S}_i\}$ of the training set $\{(\mathbf{x}_n, y_n)\}_{n=1}^{N}$,

on which its bisector (with parameters $\boldsymbol{\theta}_i$) will operate. If $i$ and $j$ are not descendants of each other, then their subsets are disjoint. Since $\mathcal{L}(\boldsymbol{\Theta})$ is a separable sum over the $N$ points, then the theorem follows, with $\mathcal{L}_i(\boldsymbol{\theta}_i)$ summing those training points in $\mathcal{S}_i$, $\mathcal{L}_j(\boldsymbol{\theta}_j)$ summing those in $\mathcal{S}_j$, and the remaining points being summed in the "constant" term. That is, the respective terms are:

$$\mathcal{L}(\boldsymbol{\Theta}) = \underbrace{\sum_{n \in \mathcal{S}_i} L(y_n, T(\mathbf{x}_n; \boldsymbol{\Theta}))}_{\mathcal{L}_i(\boldsymbol{\theta}_i)} + \underbrace{\sum_{n \in \mathcal{S}_j} L(y_n, T(\mathbf{x}_n; \boldsymbol{\Theta}))}_{\mathcal{L}_i(\boldsymbol{\theta}_j)} + \underbrace{\sum_{n \in \{1,\dots,N\} \backslash (\mathcal{S}_i \cup \mathcal{S}_j)} L(y_n, T(\mathbf{x}_n; \boldsymbol{\Theta}))}_{\text{constant}}.$$

Note that $\mathcal{L}_i$ depends on the parameters $\boldsymbol{\theta}_k$ of other nodes $k$ besides $i$ but it does not depend on $\boldsymbol{\theta}_j$. Likewise, $\mathcal{L}_j$ depends on other nodes' parameters besides $j$'s but it does not depend on $\boldsymbol{\theta}_i$; and the "constant" term depends on other nodes' parameters but it does not depend on $\boldsymbol{\theta}_i$ or $\boldsymbol{\theta}_j$. □

*The separability condition allows us to optimize separately (and in parallel) over the parameters of any set of nodes that are not descendants of each other (fixing the parameters of the remaining nodes).* This has two advantages. First, we expect a deeper decrease of the loss, because we optimize over a large set of parameters exactly. This is because optimizing over each node can be done exactly (see some caveats later) and the nodes separate. Second, the computation is fast: the joint problem over the set becomes a collection of smaller independent problems over the nodes that can be solved in parallel (if so desired). There are many possible choices of such node sets, and it is of interest to make those sets as big as possible, so that we make large, fast moves in search space. One example of set is "all nodes at the same depth" (distance from the root), and we will focus on it.

**TAO: alternating optimization over depth levels of the tree**   We cycle over depth levels from the bottom (leaves) to the top (root) and iterate bottom-top, bottom-top, etc. (i.e., reverse breadth-first order). We experimented with other orders (top-bottom, or alternating bottom-top and top-bottom) but found little difference in the results for both axis-aligned and oblique trees. At a given depth level, we optimize in parallel over all the nodes. We call this algorithm *Tree Alternating Optimization (TAO)*. The optimization over each node is described below. Before, we make some observations.

As TAO iterates, the root-leaf path followed by each training point changes and so does the set of points that reach a particular node. This can cause dead branches and pure subtrees, which we remove after convergence. *Dead branches* arise if, after optimizing over a node, some of its subtrees (a child or other descendants) become empty because they receive no training points from their parent (which sends all its points to the other child). The subtree of a node with one empty child can be replaced with the non-empty child's subtree. We do not do this as soon as they become empty in case they might become non-empty again. *Pure subtrees* arise if, after optimizing over a node, some of its subtrees become pure (i.e., all their points have the same label). A pure subtree can be replaced with a leaf but, as with dead branches, we do this after convergence, in case they become impure again. During each pass, only non-empty, impure nodes are processed, so dead branches and pure subtrees are ignored, which accelerates the algorithm. *This means that TAO can actually modify the tree structure, by reducing the size of the tree*; we call this *indirect pruning*, and it is very significant with sparse oblique trees (described later). It is a good thing because we achieve (while always decreasing the training loss) a smaller tree that is faster, takes less space and—having fewer parameters—probably generalizes better. TAO pseudocode appears in the supplementary material.

**Optimizing the misclassification error at a single node**   As we show below, *optimizing eq. (2), the $K$-class misclassification error of the tree over a node's parameters (decision function at an internal node or predictor model at a leaf), reduces to optimizing a misclassification error of a simpler classifier.* In some important special cases this "reduced problem" can be solved exactly, but in general it is an NP-hard problem [17, 21]. In the latter case, we can approximate it by a surrogate loss (e.g. logistic or hinge loss with a support vector machine). We consider each type of node next (leaf or internal).

Optimizing (2) over a leaf is equivalent to minimizing the $K$-class misclassification error over the subset of training points that reach the leaf. If the classifier at the leaf is a constant label, this is solved exactly by majority vote (i.e., setting the leaf label to the most frequent label in the leaf's subset of points). If the classifier at the leaf is some other model, we can train it using a surrogate of the misclassification error.

Optimizing (2) over an internal node $i$ is exactly equivalent to a *reduced problem*: a *binary* misclassification loss for a certain subset $\mathcal{C}_i$ (defined below) of the training points over the parameters $\boldsymbol{\theta}_i$ of $i$'s decision function $f_i(\mathbf{x}; \boldsymbol{\theta}_i)$. The argument is subtle; we show it step by step. Firstly, optimizing the misclassification error over $\boldsymbol{\theta}_i$ in (2), where is summed over the whole training set, is equivalent to optimizing it over the subset of training points $\mathcal{S}_i$ that reach node $i$. Next, the fate of a point $\mathbf{x}_n \in \mathcal{S}_i$ (i.e., the label the tree will predict for it) depends only on which of $i$'s children it follows, because the decision functions and leaf predictor models in the subtree rooted at $i$ are fixed (in other words, the subtree of each child of $i$ is a fixed decision tree that classifies whatever is inputted to it). Hence, call $z_j \in \{1, \ldots, K\}$ the label predicted for $\mathbf{x}_n$ if following child $j$ (where $j$ is left or right). Now, comparing the ground-truth label $y_n$ of $\mathbf{x}_n$ in the training set with the predicted label $z_j$ for child $j$, they can either be equal ($y_n = z_j$, correct classification) or not ($y_n \neq z_j$, incorrect classification). Hence, if $\mathbf{x}_n$ is correctly classified for all children $j \in \{\text{left}, \text{right}\}$, or incorrectly classified for all children $j \in \{\text{left}, \text{right}\}$, the fate for this point cannot be altered by changing the decision function at $i$, and we call such a point "don't-care". It can be removed from the loss since it contributes an additive constant. Therefore, the only points ("care" points) whose fate does depend on the parameters of $i$'s decision function are those for which $z_{\text{left}} = y_n \neq z_{\text{right}}$ or $z_{\text{right}} = y_n \neq z_{\text{left}}$. Then, we can define a new, binary classification problem over the parameters $\boldsymbol{\theta}_i$ of the decision function $f_i(\mathbf{x}; \boldsymbol{\theta}_i)$ on the "care" points $\mathcal{C}_i \subseteq \mathcal{S}_i$ where $\mathbf{x}_n \in \mathcal{C}_i$ has a label $\overline{y}_n \in \{\text{left}, \text{right}\}$, according to which child classifies it correctly. The misclassification error in this problem equals the misclassification error in eq. (2) for each "care" point. In summary, we have proven the following theorem.

**Theorem 3.2** (reduced problem). *Let $T(\mathbf{x}; \boldsymbol{\Theta})$ be the predictive function of a rooted directed binary decision tree and $i$ be any internal node in the tree with decision function $f_i(\mathbf{x}; \boldsymbol{\theta}_i)$. The tree's $K$-class misclassification error* (2) *can be written as:*

$$\mathcal{L}(\boldsymbol{\Theta}) = \sum_{n=1}^{N} L(y_n, T(\mathbf{x}_n; \boldsymbol{\Theta})) = \sum_{n \in \mathcal{C}_i} L(\overline{y}_n, f_i(\mathbf{x}_n; \boldsymbol{\theta}_i)) + constant \tag{3}$$

*where the "constant" does not depend on $\boldsymbol{\theta}_i$, $\mathcal{C}_i \subset \{1, \ldots, N\}$ is the set of "care" training points for $i$ defined above, and $\overline{y}_n \in \{\text{left}, \text{right}\}$ is the child that leads to a correct classification for $\mathbf{x}_n$ under $i$'s current subtree.*

All is left now is how to solve this binary misclassification loss problem:

$$\text{Reduced problem:} \qquad \min_{\boldsymbol{\theta}_i} \sum_{n \in \mathcal{C}_i} L(\overline{y}_n, f_i(\mathbf{x}_n; \boldsymbol{\theta}_i)). \tag{4}$$

For axis-aligned trees, it can be solved exactly by enumeration over features and splits, just as in the CART algorithm to minimize the impurity. For oblique trees, we approximate it by a surrogate loss.

The reduced problem is, of course, much easier to solve than the original loss over the tree. In particular for the oblique case (where the node decision function is a hyperplane, for which enumeration is intractable), the reduced problem introduces a crucial advantage over the traditional impurity minimization in CART. The latter is a non-differentiable, unsupervised problem, which is solved rather inaccurately via coordinate descent over the hyperplane weights. The reduced problem is non-differentiable but supervised and can be conveniently approximated by a surrogate binary classification loss, much easier to solve accurately. This improved optimization translates into much better trees using TAO, as seen in our experiments.

**Sparse oblique trees** The equivalence of optimizing (2) over one oblique node to the reduced problem (4) makes it computationally easy to introduce constraints over the weight vector and hence learn more flexible types of oblique trees than was possible before. In this paper we propose to learn oblique nodes involving few features. We can do this by adding an $\ell_1$ penalty "$\lambda \sum_{\text{nodes } i} \|\mathbf{w}_i\|_1$" to the misclassification error (2) where $\lambda \geq 0$ controls the sparsity: from no sparsity for $\lambda = 0$ to all weight vectors in all nodes being zero if $\lambda$ is large enough. Since this penalty separates over nodes, the only change in TAO is that the optimization over node $i$ in eq. (4) has a penalty "$\lambda\|\mathbf{w}_i\|_1$". This can be easily combined with the formulation above, resulting in an $\ell_1$-regularized linear SVM or logistic regression [19, sections 3.2 and 3.6], a convex problem for which well-developed code exists, such as LIBLINEAR [15]. Alternatively, one can use an $\ell_0$ penalty or constraint on the weights, for which good optimization algorithms also exist [14].

**Convergence and computational complexity**   Optimizing the misclassification loss $\mathcal{L}$ is NP-hard in general [17, 21, 26] and we have no approximation guarantees for TAO at present. TAO does converge to a local optimum in the sense of alternating optimization (as in $k$-means), i.e., when no more progress can be made by optimizing one subset of nodes given the rest. For oblique trees, the complexity of one TAO iteration (pass over all nodes) is upper bounded by the tree depth times the cost of solving an SVM on the whole training set, and is typically quite smaller than that. For axis-aligned trees, one TAO iteration is comparable to running CART to grow a tree of the same size, since in both cases the nodes run an enumeration procedure over features and thresholds. See details in the supplementary material.

# 4   Experiments: sparse oblique trees for MNIST digits

The supplementary material gives additional experiments using TAO to optimize axis-aligned and oblique trees on over 10 datasets and comparing with other methods for optimizing trees. In a nutshell, TAO produces trees that significantly improve over the CART baseline for axis-aligned and especially oblique trees, and also consistently beat the other methods. But where TAO is truly remarkable is with sparse oblique trees, and we explore this here in the MNIST benchmark [27].

We induce an initial tree for TAO using the CART algorithm[1] (greedy growing and pruning) either for axis-aligned trees (enumeration over features/thresholds) or oblique trees (coordinate descent over weights, picking the best of several random restarts [28]). The node optimization uses an $\ell_1$-regularized linear SVM with slack hyperparameter $C \geq 0$ (so the TAO sparsity hyperparameter in section 3 is $\lambda = 1/C$), implemented with LIBLINEAR [15]. The rest of our code is in Matlab. We stop TAO when the training misclassification loss decreases but by less than 0.5%, or the number of iterations (passes over all nodes) reaches 14 (in practice TAO stops after around 7 iterations).

Sparse oblique trees behave like the LASSO [20]: we have a path of trees as a function of the sparsity hyperparameter $C$, from $\infty$ (no $\ell_1$ penalty) to 0 (all parameters zero). We estimate this path by inducing an initial CART tree and running TAO for a sequence of decreasing $C$ values, where the tree at the current $C$ value is used to initialize TAO for the next, smaller $C$ value. We constructed paths using initial CART trees of depths 4 to 12 (both axis-aligned and oblique) on the MNIST dataset, splitting its training set of 60k points into 48k training and 12k validation (to determine an optimal $C$ or depth), and reporting generalization error on its 10k test points. The training time for each $C$ value is roughly between 1 minute (for depth 4) and 4 minutes (for depth 12).

**Path of trees**   The resulting paths are best viewed in our supplementary animations. Fig. 1 shows three representative trees from the path for depth 12: the initial CART tree (which was oblique), the tree with optimal validation error and an oversparse tree. It also plots various measures as a function of $C$. Several observations are obvious, as follows.

As soon as TAO runs on the initial CART tree (for the largest $C$ value, which imposes little sparsity), the improvement is drastic: from a training/test error of 1.95%/11.03% down to 0.09%/5.66%. The tree is pruned from 410 to 230 internal nodes (the number of leaves for a binary tree equals the number of internal nodes plus 1). The TAO tree is more balanced: the samples are distributed more evenly over the tree branches and the training subset that a node receives is more pure. This can be seen explicitly from the node histograms (tree as binary heap animations in the supplementary material) or indirectly from the sample mean of the node.

Further decreasing $C$ imposes more sparsity and this leads to progressive pruning of the tree and ever sparser weight vectors at the internal nodes. The large changes in topology are caused by postprocessing the tree to remove dead branches. The number of nonzero weights and the number of nodes (internal and leaves) decreases monotonically with $C$. The training error slowly increases but the test error remains about constant. In general, depending on the initial tree size, we find the validation error (not shown) and the test error are minimal for some range of $C$); trees there will generalize best. Further decreasing $C$ (beyond 0.01 in the figure) increases both training and test error and produces smaller trees with sparser weight vectors that underfit.

**Inference runtime** For inference (prediction), each point follows a different root-leaf path. We report its mean/min/max path length (number of nodes) and runtime (number of operations, here scalar multiplications) over the training set, for each $C$. It decreases mostly monotonically with $C$. The inference time is extremely fast because the root-leaf path involves a handful of nodes and each node's decision function looks at a few pixels of the image. This is orders-of-magnitude faster than kernel machines, deep nets, random forests or nearest-neighbor classifiers, and is a major advantage of trees. The same can be said of the model storage required. This is especially important at present given the need to deploy classifiers on resource-constrained IoT devices [9–11, 18, 24].

**Classification accuracy** The best test error for the TAO trees we obtained (having initial depth up to 12) is around 5%. To put this in perspective, we plot the error reported for MNIST for a range of models [27] vs. the number of parameters in fig. 1. The tree error is much better than than of linear models ($\approx$12%) and boosted stumps (7.7%) and is comparable to that of a 2-layer neural net and a 3-nearest-neighbor classifier. Better errors can of course be achieved by many-parameter, complex models such as kernel SVMs or convolutional nets (not shown), or using image-specific transformations and features. Our trees operate directly on the image pixels with no special transformation and are astoundingly small and fast given their accuracy. For example, our tree achieves about the same test error as a 3-nearest-neighbor classifier, but the tree compares the input image with at most $\approx$6 sparse "images" (weight vectors), rather than with the 60 000 dense training images.

**Interpretable trees** By using a high sparsity penalty, TAO allows us to obtain trees of suboptimal but reasonable test error that have a really small number of nodes and nonzero weights and are eminently interpretable. Fig. 1 shows an example for $C = 0.01$ (test error 10.19%, 0.22% nonzeros, 17 leaves). Examining the nodes' weight vectors shows that the few weights that are not zero are strategically located to discriminate between specific classes or groups of classes, and essentially detect patterns characterized by the presence or absence of strokes in certain locations. Nodes use minimal features to separate large groups of digits, and leaf parents often separate very similar digits that differ on just one or two strokes. We mention some examples (referring to the nodes by their index in the figure). Node 53 separates 3s from 5s by detecting the small vertical stroke that precisely differentiates these digits (blue = negative, red = positive). Node 5 separates 4s from 9s by detecting the presence of a top horizontal stroke. Node 2 separates 7s from {4,9} by detecting ink in the left-middle of the image. Node 6 separates 0s from {1,2,3,5,8} by detecting the presence of ink in the center of the image but not in its sides. Node 1 (the root) separates {4,7,9} from the remaining digits. Also, once the tree is sparse enough, several of these weight vectors (such as nodes 2 and 5) tend to appear in the tree regardless of the initial tree and depth (see supplementary animations).

In a sense, each decision node pays attention to a simple but high-level concept so the tree classifies digits by asking a few conceptual questions about the relative spatial presence of strokes or ink. A root-leaf path can then be seen as a sequence of conceptual questions that "define" a class. This is very different from the way convolutional neural networks operate, by constructing a large number of "features" that start very simple (e.g. edge detectors) and are combined into progressively more abstract features. While deep neural nets get very accurate predictions (so they are able to classify correctly even unusual digit shapes), this is achieved by very complex models that are not easy to interpret. Our trees do not reach such high accuracy, but arguably they are able to learn the more high-level, conceptual structure of each digit class.

## 5   Experiments: comparison with forest-based nearest-neighbor classifiers

As requested by a reviewer, we compared CART+TAO with fast, forest-based algorithms that approximate a nearest-neighbor classifier (see [25] and references therein). Roughly speaking, these algorithms construct a tree that can approximate a nearest neighbor search and have a controllable tradeoff between approximation error and search speed. Thus, they can be used to approximate a nearest-neighbor classifier fast. On top of that, they can be ensembled into a forest. Although the comparison is not apples-to-apples (since the latter classifiers are not of the decision-tree type, and also are forests rather than a single tree), it is still very interesting.

We followed the protocol of [25], which lists results for cover trees (CT) [5], random kd-trees (forest of 4 trees) [16] and boundary forest (BF) (50 trees) [25]. Table 1 shows our results for TAO on oblique trees (we initialized TAO from both axis-aligned and oblique CART trees and picked the

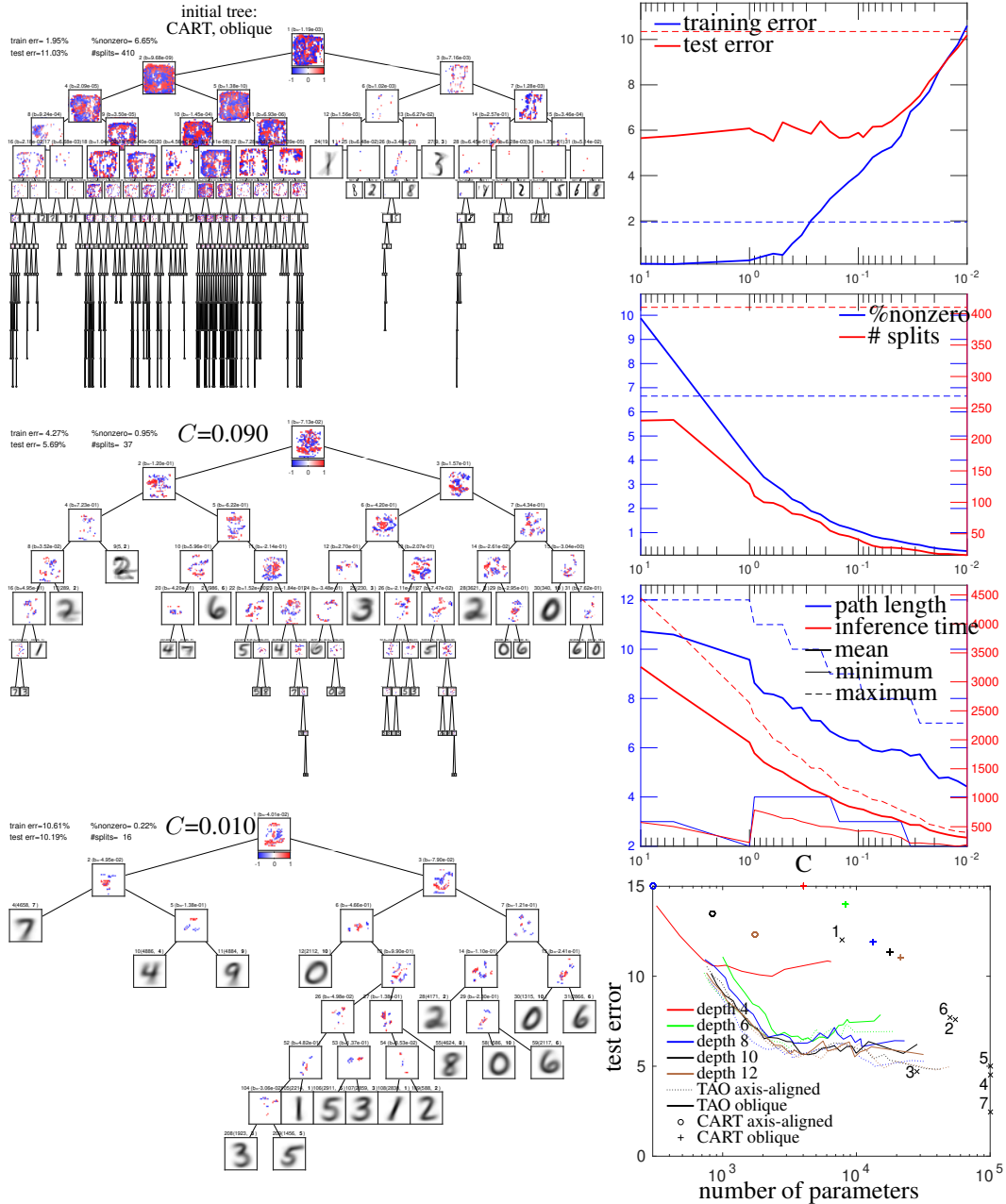

Figure 1: Sparse oblique trees for MNIST. *Left plots*: initial CART tree and sparse oblique trees for $C = 0.09$ and 0.01. For each internal node, we show its index and bias value and plot its weight vector (red = positive, blue = negative, white = zero); you may need to zoom into the image. For each leaf, we plot the mean of its training points and show something like "4(4658,**7**)" where 4 is its index, 4658 is the number of training points it receives, and **7** is its digit class. *Right plots*: several measures of the tree as a function of $C \geq 0$: training/test error; proportion of nonzero weights and number of internal nodes; and length of root-leaf path and inference time (in scalar multiplications) for an input sample. The bottom plot shows test error vs. number of parameters for sparse oblique trees of different depths (color-coded), initialized from a CART tree that is either axis-aligned (dotted line) or oblique (solid line). The markers correspond to the initial CART trees ($\circ$, $+$) or to models from [27], numbered as follows: 1) linear classifiers, 2) one-vs-all linear classifiers, 3) 2-layer neural net with 300 hidden units, 4) 2-layer neural net with 1 000 hidden units, 5) 3-nearest-neighbor classifier, 6) one-vs-all classifiers where each classifier consists of 50 000 boosted decision stumps (each operating over a feature and threshold), 7) 3-layer neural net with 500+100 hidden units. Values outside the axes limits are projected on the boundary of the plots.

Table 1: Comparison with forest-based algorithms that approximate a nearest-neighbor classifier.

| dataset ($N \times D$, $K$) | Test error (%) | | | | Inference time on entire test set (seconds) | | | | $C$ |
|---|---|---|---|---|---|---|---|---|---|
| | TAO | BF | R-kd | CT | TAO | BF | R-kd | CT | TAO |
| MNIST (60 000×784, 10) | 5.69 | 2.24 | 3.08 | 2.99 | 0.18 | 23.90 | 89.20 | 417.60 | 0.09 |
| letter (10 500×16, 26) | 7.94 | 5.40 | 5.50 | 5.60 | 0.05 | 1.16 | 1.67 | 0.91 | 9.11 |
| pendigits (7 494×16, 10) | 3.14 | 2.62 | 2.26 | 2.80 | 0.01 | 0.34 | 0.75 | 0.02 | 0.03 |
| protein (17 766×357, 3) | 31.70 | 44.20 | 53.60 | 52.00 | 0.05 | 35.47 | 11.50 | 51.40 | 0.14 |
| seismic (78 823×50, 3) | 27.81 | 40.60 | 30.80 | 38.90 | 0.09 | 16.20 | 65.70 | 172.5 | 3.28 |

best result), ran on a laptop with 2 core i5 CPUs and 12GB RAM (pretty similar to the system of [25]). TAO's test error is somewhat bigger (first 3 datasets) or quite smaller (last 2 datasets) than other forest classifiers, but it always has faster inference time by at least one order of magnitude. We reiterate that *TAO produces a single tree with sparse decision nodes*.

## 6 Discussion

The way TAO works is very simple: *TAO repeatedly trains a simple classifier (binary at the decision nodes, $K$-class at the leaves) while monotonically decreasing the objective function*. The only thing that changes over iterations is the subset of training instances on which each classifier is trained. In order to optimize the misclassification error, TAO fundamentally relies on alternating optimization. This is most effective when two circumstances apply. 1) Some separability into blocks exists in the problem, as e.g. in matrix factorization, or is created via auxiliary variables, as e.g. with consensus problems [6] or nested functions [12]. And 2) the step over each block is easy and ideally exact. All this applies here thanks to the separability condition and the reduced problem.

Two important remarks. First, note TAO is very different from coordinate descent in CART [8, 28]. The latter optimizes the impurity of a single node; each step updates a single weight of its hyperplane. TAO optimizes the misclassification error of the entire tree; each step updates one entire set of nodes (i.e., all the weights of all the hyperplanes in those nodes). Second, what we really want to minimize is the misclassification error on the data, not the impurity in each node. The latter, while useful to construct a good tree structure and initial node parameters, is only indirectly related to the classification accuracy.

The quality of the TAO result naturally depends on the initial tree it is run on. A good strategy appears to be to grow a large tree with CART that overfits the data (or a large tree with random parameters) and let TAO prune it, particularly if using a sparsity penalty with oblique trees. TAO also depends on the choice of surrogate loss in the node (decision or leaf) optimization. In our experience with the logistic or hinge loss the TAO trees considerably improve over the initial CART or random tree.

## 7 Conclusion

We have presented Tree Alternating Optimization (TAO), a scalable algorithm that can find a local optimum of oblique trees given a fixed structure, in the sense of repeatedly decreasing the misclassification loss until no more progress can be done. A critical difference with the standard tree induction algorithm is that we do not optimize a proxy measure (the impurity) greedily one node at a time, but the misclassification error itself, jointly and iteratively over all nodes. We suggest to use TAO as postprocessing after the usual greedy tree induction in CART, or to run TAO directly on a random initial tree.

TAO could make oblique trees widespread in practice and replace to some extent the considerably less flexible axis-aligned trees. Even more interesting are the sparse oblique trees we propose. These can strike a good compromise between flexible modeling of features (involving complex local correlations, as with image data) and using few features in each node, hence producing a relatively accurate tree that is very small, fast and interpretable. For MNIST, we believe this is the first time that a single decision tree achieves such high accuracy, comparable to that of much larger models.

Our work opens up important extensions, among others: to other types of trees, such as regression trees; to other types of nodes beyond linear bisectors or constant-class leaves; to ensembles of trees; to using TAO with a search over tree structures; and to combining trees with other models.

**Acknowledgements**

Work funded in part by NSF award IIS–1423515.

## Footnotes

[1]During the review period we found out that TAO performs about as well on random initial trees (having random parameters at the nodes) as on trees induced by CART. This would make TAO a stand-alone learning algorithm rather than a postprocessing step over a CART tree. We will report on this in a separate publication.

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
