[Supplementary Material]

<p style="text-align:center">Supplementary material for:<br>
Alternating Optimization of Decision Trees,<br>
with Application to Learning Sparse Oblique Trees</p>

Miguel Á. Carreira-Perpiñán      Pooya Tavallali

Electrical Engineering and Computer Science, University of California, Merced

`http://eecs.ucmerced.edu`

October 26, 2018

**Abstract**

This is supplementary material for the "main paper" (Carreira-Perpiñán and Tavallali, 2018). We provide: pseudocode for the TAO algorithm; details on the convergence and computational complexity; additional experiments.

# 1   Tree Alternating Optimization (TAO)

Fig. 1 shows pseudocode for the TAO algorithm.

```
input training set {(xₙ, yₙ)}ⁿ₌₁ᴺ ⊂ ℝᴰ × {1, …, K}
        initial tree T (from CART, or random)
repeat
    for i ∈ nodes of T, visited in reverse BFS
        if i is a leaf then
            y_i ← majority label of the training points that reach i
        else
            θ_i ← minimizer of the reduced problem, eq. (4)
until stop
postprocess T: remove dead branches & pure subtrees
return T
```

Figure 1: Pseudocode for the tree alternating optimization (TAO) algorithm, assuming a constant label at the leaves. Visiting each node in reverse breadth-first search (BFS) order means scanning depths from $\text{depth}(T)$ down to 1, and at each depth processing (in parallel, if so desired) all nodes at that depth. "stop" occurs when either the objective function decreases less than a set tolerance or the number of iterations reaches a set limit.

## 1.1 Convergence

Optimizing the misclassification loss $\mathcal{L}$ is NP-hard in general (Megiddo, 1988; Hoffgen et al., 1995; Guruswami and Raghavendra, 2009) and we have no approximation guarantees for TAO at present. That said, TAO performs very well in our experiments, drastically improving over CART and other approaches. We discuss here convergence in the sense of alternating optimization (as in $k$-means), i.e., when no more progress can be made by optimizing one subset of nodes given the rest.

If the optimization of the misclassification error over each node is exact (e.g. for axis-aligned trees with a constant label at each leaf) then TAO will monotonically decrease $\mathcal{L}$ and stop after a finite number of iterations (passes over all nodes of the tree), when none of the nodes' parameters change. This is because 1) the search space is finite, given by all the possible assignments of training points to leaves (in fact, the tree can be seen as a mechanism to assign each point to a leaf); and 2) for each such assignment encountered during TAO iterations we solve the optimization over the node parameters exactly. (Strictly speaking, this ignores the possibility of cycling in parameter space without reducing the loss.)

If the optimization of the misclassification error over each node is approximated via a surrogate loss, as with oblique trees, the solution is inexact and can even increase the misclassification loss $\mathcal{L}$ over that of the current parameters. We do observe this as tiny oscillations in the value $\mathcal{L}$ when TAO is converging (since it is then harder to improve $\mathcal{L}$ via the surrogate). We can always avoid this by updating the parameters only if $\mathcal{L}$ decreases, in which case TAO will monotonically decrease $\mathcal{L}$ and stop after a finite number of iterations as before. However, experimentally we find slightly better trees by simply ignoring these tiny oscillations and continue training, in effect letting the surrogate drive the decision functions. We then stop either when we reach a maximum number of iterations or when the parameters do not change anymore. In our experiments, we find TAO needs around 7 iterations to achieve best results. All these arguments apply when we optimize the loss plus an $\ell_1$ penalty for sparsity.

## 1.2 Computational complexity

For oblique trees, the complexity of one TAO iteration (pass over all nodes) is upper bounded by the tree depth times the cost of solving an SVM on the whole training set, and is typically quite smaller than that. Let us see this. Consider all the nodes $i$ at a given depth. Each node solves a reduced problem on a "care" subset $\mathcal{C}_i$ of the training set, of size $N_i$. The subsets $\mathcal{C}_i$ from these nodes are disjoint so their aggregated size is at most $N$, and typically quite less (because of removing "don't care" points, or because the tree is not complete so there are missing nodes at that depth). Each node solves an SVM on its subset $\mathcal{C}_i$ in time $\mathcal{O}(DN^\alpha)$ for $\alpha \geq 1$ (exactly what this is depends on the SVM solver; Bottou and Lin, 2007, section 4.2). Hence, since $\sum_i N_i^\alpha \leq \left(\sum_i N_i\right)^\alpha \leq N^\alpha$ if $\alpha \geq 1$, solving all the SVMs at the same depth is at most as costly as solving a single SVM on all $N$ points. The overhead of propagating points through the tree to determine the subsets is negligible compared to this.

For axis-aligned trees, the complexity of one TAO iteration (pass over all nodes) is comparable to that of running CART to grow a tree of the same size. This is because the enumeration procedure over features and thresholds is essentially the same (except that we evaluate the binary loss of the reduced problem rather than the impurity), and can be done efficiently via incremental computation.

# 2 Additional experiments

We induce an initial tree for TAO using the CART algorithm (greedy growing and pruning) either for axis-aligned trees (enumeration over features/thresholds) or oblique trees (coordinate descent over weights, picking the best of several random restarts as suggested by Murthy et al., 1994). The node optimization in oblique trees uses as surrogate an $\ell_2$-regularized linear SVM with slack hyperparameter $C = 1$. For sparse oblique trees, we use an $\ell_1$-regularized linear SVM with slack penalty hyperparameter $C \geq 0$, so the TAO sparsity hyperparameter is $\lambda = 1/C$. Both are implemented with LIBLINEAR (Fan et al., 2008). The rest of our code is in Matlab. In this set of experiments, we used the following stopping criterion for TAO. With axis-aligned trees, TAO stops when the parameters do not change (2–4 iterations in practice). With (sparse) oblique trees, we stop TAO when the misclassification loss in the training set decreases but by less than 0.5%, or the number of iterations (passes over all nodes) reaches 14 (in practice TAO stops after around 7 iterations).

## 2.1 Axis-aligned trees

Bertsimas and Dunn (2017) have recently advocated the use of mixed-integer optimization (based on branch-and-bound) to find the globally optimum tree in their OCT algorithm. Because of its worst-case exponential cost, they stop OCT after 2 hours at most and return the best tree found thus far. We follow their experimental setup (data partition into 50% training, 25% validation, 25% test) on several of their datasets (from Lichman, 2013). For CART and TAO we use 10 random partitions and report mean and standard deviation training and test accuracy. For OCT we report the single test accuracy provided by Bertsimas and Dunn (2017). (We cannot replicate their experiments because the OCT code is not available online, and relies on a commercial MIO solver.) Table 1 shows the results. In all cases, TAO has the best training and test accuracy, sometimes by a considerable margin. Its runtime was less than 0.05 seconds on all datasets.

| dataset | depth | OCT | CART | TAO |
|---|---|---|---|---|
| Balance | 2 | (67.1) | 70.0±2.7 (63.9±2.7) | **72.5**±1.6 (**69.5**±2.9) |
| scale | 3 | (68.9) | 75.8±1.4 (71.1±1.4) | **76.9**±1.1 (**71.6**±1.9) |
| $(625 \times 4, 3)$ | 4 | (71.6) | 81.0±2.3 (77.9±2.3) | **84.0**±1.5 (**79.8**±3.1) |
| Banknote | 2 | (90.1) | 90.3±0.3 (88.9±0.3) | **91.9**±0.4 (**90.6**±0.9) |
| authentication | 3 | (89.6) | 95.0±1.3 (93.9±1.3) | **96.0**±0.5 (**95.7**±1.2) |
| $(1372 \times 4, 2)$ | 4 | (90.7) | 97.7±0.8 (96.2±0.8) | **98.9**±0.7 (**97.2**±0.7) |
| Blood | 2 | (75.5) | 76.0±0.4 (75.2±0.9) | **78.0**±0.8 (**75.8**±2.0) |
| transfusion | 3 | (**77.0**) | 79.0±1.2 (76.7±1.2) | **79.5**±1.0 (**77.0**±2.0) |
| $(748 \times 4, 2)$ | 4 | (77.0) | 80.0±1.1 (76.6±1.1) | **81.6**±1.3 (**77.2**±1.3) |
| Breast | 2 | (91.9) | 93.5±1.3 (91.0±1.3) | **95.0**±0.5 (**92.7**±2.2) |
| cancer-diagnostic | 3 | (91.5) | 95.0±0.6 (93.0±0.6) | **97.0**±0.6 (**93.1**±1.4) |
| $(569 \times 30, 2)$ | 4 | (91.5) | 97.4±0.7 (93.0±0.7) | **98.0**±0.5 (**93.2**±0.5) |
| Spambase | 2 | (84.3) | 83.5±2.1 (83.6±2.5) | **86.5**±0.7 (**86.1**±1.0) |
| | 3 | (86.0) | 87.8±0.7 (86.9±1.3) | **90.0**±0.4 (**89.1**±1.0) |
| $(4601 \times 57, 2)$ | 4 | (86.1) | 90.7±0.5 (89.5±1.0) | **91.8**±0.3 (**90.3**±0.8) |

Table 1: Axis-aligned trees: mean ± stdev training (test) classification accuracy for different datasets (sample size × dimensionality, # classes), tree depths and optimization methods.

## 2.2   Oblique trees

We run TAO on an axis-aligned tree induced by CART. We use datasets (without applying any normalization ot them) as in Norouzi et al. (2015b) and follow their experimental setup to partition the data into training, validation and test. We compare with CART (axis-aligned and oblique) and with the $CO_2$ algorithm of Norouzi et al. (2015b,a) (two versions, greedy and non-greedy). Figure 2 shows the results (note that the green and brown validation curves for the $CO_2$ algorithms are missing, because they were not provided in the original papers). Again, with very few exceptions, TAO beats all methods, often by a large margin.

Because TAO does a better job at optimizing the misclassification loss, it can overfit faster (as a function of the tree depth) than other algorithms. This can be seen comparing the TAO curves for the training accuracy, which always increase, with those for validation and test accuracy, which increase and then eventually flatten or decrease slightly. This is not a problem with TAO, which is doing its job well. It is a model selection problem, which can be controlled with cross-validation to select the best tree size (namely, the smallest tree size that achieves about best validation accuracy).

Figure 2: Oblique trees: training and test classification accuracy for different datasets (sample size ×
dimensionality, # classes), tree depths and optimization methods.

## 2.3 Sparse oblique trees in MNIST

Fig. 3 shows the test error of TAO trees and various models on MNIST vs. two model size measures: inference runtime in number of operations (scalar multiplications) and number of parameters. For all models shown other than trees, the number of scalar multiplications at inference is roughly equal to the number of parameters. For trees, inference requires following a single root-leaf path (dependent on the input instance), which involves a small subset of the parameters of the tree, so the number of scalar multiplications at inference is much smaller than the number of parameters; we report the mean over all training instances.

| model from `http://yann.lecun.com/exdb/mnist` | number of parameters | test error |
|---|---|---|
| 1 linear classifier | 7 850 | 12.0 |
| 2 one-vs-all linear classifiers | 55 280 | 7.6 |
| 3 2-layer neural net with 300 hidden units | 28 200 | 4.7 |
| 4 2-layer neural net with 1 000 hidden units | 794 000 | 4.5 |
| 5 3-nearest-neighbor classifier | 47 040 000 | 5.0 |
| 6 one-vs-all classifiers where each classifier consists of 50 000 boosted decision stumps (each operating over a feature and threshold) | 50 000 | 7.7 |
| 7 3-layer neural net with 500+100 hidden units | 443 610 | 2.5 |

Figure 3: Test error vs. model size measures for different models on MNIST. *Top left*: inference runtime in number of operations (scalar multiplications). *Top right*: number of parameters. The curves correspond to sparse oblique trees of different depths (color-coded), initialized from a CART tree that is either axis-aligned (dotted line) or oblique (solid line). The markers correspond to the initial CART trees (∘, +) or to other models labeled 1–7 (see table). Values outside the axes limits are projected on the boundary of the plots. *Bottom table*: models from `http://yann.lecun.com/exdb/mnist` and their numeric values for number of operations (scalar multiplications) and number of parameters.