[Reviews · NeurIPS 2018]

Reviewer 1



I have red the author's response. Thank you for doing an extra set of experiments in such a small time frame. I will raise my rating of the paper. I still think that, give the heuristic nature of the method, more experiments are necessary. But I understand these will have to go onto a new paper, which I hope the authors publish in the future. Finally, I hope that the authors will include the new set of experiments in their rebuttal, and maybe more if they have time, on the appendix of their paper. It will only help their paper. Congrats! ----------------------------- Summary The authors propose a method to improve the decision rules at the nodes of a decision tree. Their method requires an initial decision tree. The topology of this tree will be fixed, and only the decision rules at each node will be adjusted. The idea behind the proposed adjustment is based on the observation that, fixing all of the parameters of all the nodes except the parameters of node i, the likelihood function for the whole tree reduces to the likelihood function of a simple K-classes classifier. This simple classifier can be trained efficiently (using existing techniques) and doing so will always guarantee that the overall loss will decrease when compared to the loss for the initial decision tree. Furthermore, if node i (resp. j) is not a descendant of node j (resp. i), the simple classifiers for nodes i and j can be trained in parallel. Each simpler classifier can be sparsified using l1-regularization, which facilitates deriving interpretable models. Their algorithm, TAO, systematically improves the decision rules at different nodes until the loss does not decrease substantially. After the decision rules are improved, and especially when sparsity is used, many nodes become irrelevant, which allows to effectively reduced the size of the original tree, again aiding interpretability. The method currently lacks theoretical analysis. Quality The idea proposed is simple but seems very promising. Great job ! Using TAO won’t hurt accuracy, and it might even improve accuracy. The heuristic nature of their overall algorithm implies that, as far as the potential for improving accuracy goes, and at least until theory is developed, one needs to make arguments via numerical experiments. There are two different kinds of experiments in this regard. The first type is starting with an existing decision tree learning algorithm, and showing how much their method can improve on the initial tree. The authors do these kinds of experiments. The second type of experiment is comparing their method with other methods for classification that are also heuristics and very fast, but that are not of the decision-tree type, and hence cannot be improved via their method. The authors mention comparisons with nearest-neighbor methods. Unfortunately, regarding this second kind of experiments, the authors could have done a better job. It is vital that they compare CART+ their algorithm against other non decision-tree methods more extensively. A few of suggestions are cover trees [1], k-d trees [2], and the boundary tree algorithm [3]. These algorithms can be very trained fast, and classify points equally fast, and have accuracy greater than 5% on MNIST. [1] @inproceedings{beygelzimer2006cover, title={Cover trees for nearest neighbor}, author={Beygelzimer, Alina and Kakade, Sham and Langford, John}, booktitle={Proceedings of the 23rd international conference on Machine learning}, pages={97--104}, year={2006}, organization={ACM} } [2] @article{friedman1977algorithm, title={An algorithm for finding best matches in logarithmic expected time}, author={Friedman, Jerome H and Bentley, Jon Louis and Finkel, Raphael Ari}, journal={ACM Transactions on Mathematical Software (TOMS)}, volume={3}, number={3}, pages={209--226}, year={1977}, publisher={ACM} } [3] @inproceedings{mathy2015boundary, title={The Boundary Forest Algorithm for Online Supervised and Unsupervised Learning.}, author={Mathy, Charles and Derbinsky, Nate and Bento, Jos{\'e} and Rosenthal, Jonathan and Yedidia, Jonathan S}, booktitle={AAAI}, pages={2864--2870}, year={2015} } ] Significance More numerical experiments are needed to assess how much their method can improve the existing heuristics for decision trees, and if these improved trees give a better classification accuracy than other non-decision-tree type of classification algorithms. At this point, the method is significant because (1) it cannot hurt performance, and (2) it seems, at least empirically, that it can be used to produced compact trees with simple (sparse) decision rules at the nodes. Originality The idea is new as far as I can tell. Clarity The paper is very clear and well written. Great job ! A few suggestions Line 127: do not use { . , .} for theta_i. Use instead ( . , . ). Line 132: I do not understand what theta_i \cap theta_j = \emptyset means. Line 189: using “fate” is confusing. Line 251: \lambda = 1/C should not be inside parenthesis but rather more emphasized Line 252: 0.5% of what? Line 337: it is misleading the use of “globally over all nodes”. The algorithm optimizes a global cost function. But it does so heuristically, by optimizing a few decision rules at a time.

Reviewer 2



Post-rebuttal: Thank you for performing additional experiments for the rebuttal. I am satisfied with your answers and so will maintain my score. Good job! ------------------------------- This paper proposes a method for post-processing decision trees for classification so that to improve their training loss. They key idea is the following: given a fixed decision tree from CART, nodes of the tree can be traversed bottom-up, level-by-level, where in each level, every node's decision function can be optimized separately. The node subproblems are standard classification problems, e.g. SVM, and can accommodate sparsity via regularization. This algorithmic trick is due to a straightforward yet useful separability condition stated and proved in Theorem 3.1. The implication of this result is that sparse oblique trees can be learned from a simple CART tree. Oblique trees are known to be hard to optimize, since each node represents a hyperplane rather than a simple split on a single feature. Experimentally, the authors show the merit of the proposed method, TAO, on the MNIST dataset, while the appendix includes additional results on other datasets. TAO is computationally efficient and capable of substantially improving the accuracy of a vanilla CART tree. When emphasizing sparsity, the nodes' hyperplanes become more interpretable. Overall, I think this is a very good paper. The paper is generally well-written, the problem of improving a learned decision tree is well-posed, the methodology is simple and elegant, and the experimental results are convincing. I have some minor questions/suggestions below. - Extensions: in line 32, you mention possibly extensions; can you discuss this point in more detail, e.g. non-binary trees, regression? - Overfitting: in appendix 2.2, you mention TAO overfitting and a remedy using cross-validation. Please do perform that experiment so that we can be confident that TAO generalizes well. Minor comments: - Figure 1: in the bottom-right plot, the numbers for the 7 classifiers overlap, please use clearer markers. Also, the projection onto the boundaries of the plot can be misleading; maybe consider including the exact values in a table in the appendix, for all of those 7 classifiers. - Line 105: about the "exponential cost", while this is correct in theory in the worst-case, it is often not the case in practice. - Line 107: about the strict inequalities and the small constant, I don't believe this alters the feasible region if epsilon is chosen intelligently, as is done in [4]. Minor typos: - Line 183: if a constant -> is a constant

Reviewer 3



I am satisfied with the authors' response, the new experiments make a good addition and improve the paper. Thus I maintain my score and reommend accepting this work. ----- This paper presents an improvement scheme for (binary) decision trees, which takes a given tree (e.g., output of the CART algorithm) and iteratively refines it using an alternating misclassification error minimization scheme that optimizes over subsets of the tree nodes while keeping the rest fixed. The proposed method can handle both axis-aligned trees (single-feature decision nodes) as well as oblique trees (with decision hyperplanes), and in the latter case, incorporate sparsity-promoting penalties in order to obtain more interpretable oblique trees where each decision node involves a few features only. The TAO (tree alternating optimization) algorithm exploits natural separability of classification paths to define its subproblems, and although no substantial theoretical guarantees are provided for successful improvement, numerical experiments validate the approach and demonstrate that the overall classification accuracy can be improved significantly by running TAO on given trees. Based on my detailed reading (see comments below), I recommend accepting this submission. The main strength of the paper is the relatively versatile algorithmic scheme to refine decision trees that at least works well in practice, although unfortunately no strong theoretical results have been proven yet. The separability of misclassification errors over tree nodes is not a deep result, but is used cleverly to define error-minimization subproblems involving only parameters of disjoint subsets of the tree nodes that can be (approximately) solved efficiently when the remaining parameters are kept fixed, thus enabling the use of alternating optimization to improve the overall tree. All in all, the paper is well-written and the problems under consideration are well-motivated. A weak spot of the present work is the lack of theoretical understanding of the proposed method; however, as the tackled problems are NP-hard and inherently discrete (w.r.t. the tree structure/design itself), it is difficult to analyze something like approximation properties of alternating minimization schemes that are more commonly employed in continuous problems, and in light of the convincing numerical results, I do not think this is a big problem (but it should certainly be on the agenda for future research). Moving on, I would like to see some aspects clarified: 1. At some points (particularly, in the abstract), it sounds like TAO is a stand-alone algorithm to learn decision trees that „outperforms various other algorithms“ – this is misleading, as TAO requires a tree as input and cannot start from scratch by itself. Thus, please rephrase to more accurately emphasize that TAO can (significantly) improve the classification accuracy for given decision trees, and particularly given its short runtime, suggests itself as a kind of postprocessing routine for whatever method may be used to obtain a tree in the first place. 2. I would have liked to see experiments combining TAO with the MIP approaches (OCT for axis-aligned, and OCT-H for oblique trees) by Bertsimas & Dunn (cf. suppl. mat.). Specifically, what are the improvements achievable by running TAO on the trees produced by these methods, and vice versa, what may be gained by providing TAO-trees (e.g., obtained by refining CART trees) to the MIP solver as initial solutions? However, I am not sure if the OCT code is accessible, so the authors may not be able to conduct such experiments in the short time until their response is due; but perhaps this is something to consider in the future. 3. There is some lack of (notational) clarity or imprecise wording that I would like to see resolved: a) top of p. 2, in „Finding an optimal decision tree is NP-complete [...]“ – only decision problems can be contained in NP (and thus, be NP-complete if they are at the same time NP-hard), not optimization problems. So, here, it should say „... is NP-hard ...“. b) p. 2, 2nd bullet: „...tends to get stuck in poor local optima...“ (I suggest inserting „local“ to be entirely clear here) c) p. 3, 2nd paragraph: „(multilinear) linear programming“ – unclear, what „multilinear“ is to mean here; please clarify. Also, in that paragraph, the papers cited w.r.t. the claim that the „linear program is so large that the procedure is only practical for very small trees“ are over 20 years old – are you sure that this statement still holds true, given the immense progress in LP-solvers over the past two decades? It would be nice to see a more recent reference for this claim, or else a remark on own computational experience in that regard. Finally, I don't quite see why the Norouzi et al. method should not be applicable to axis-aligned trees – are they not special cases of oblique trees (with unit vectors as decision hyperplanes)? Could you please clarify why it is supposedly not applicable? (Also, starting here, you talk about „induced trees“ several times, but never clearly say what „induced“ means, so maybe define that explicitly at first occurrence.) d) p. 3, 3rd par. and 2nd bullet: Early on (on p. 1), you state the node decision question in oblique trees as strict hyperplane-inequalities, and on p. 3 write that Bertsimas & Dunn „...smooth the strict inequalities ... by adding a small constant value to one side, which can alter the solution“ – does that really matter/is this even necessary? After all, as you yourself do later on p. 3, the decision question could simply be „flipped“ so as to work with non-strict inequalities. So, please, clarify if this makes any differences and if not, make the exposition of these aspects a bit clearer and more consistent. e) In Eq. (2) should be a minimization problem (and is indeed referred to as an optimization problem later on), but the „min“ is missing. f) Thm. 3.1 and its proof: The functional $\mathcal{L}_i(\theta_i)$ should be explicitly given somewhere, preferably in the theorem or else in its proof. (Gathering what it should look like from the present description only, I believe it would be $\mathcal{L}_i(\theta_i) = \sum_{n\in\mathcal{S}_i} L(y_n, T(x_n,\{ \theta_i : i\in\mathcal{S}_i \} )$ – is that correct?) Also, the word „constant“ is a bit misleading, although it becomes clear what is meant; if possible, perhaps clarify further. g) p. 4, 4th par.: Could you please elaborate a bit what you mean by „... a smaller tree that … probably generalizes better“ ? h) p. 4, „Optimizing (2) over a leaf“: Perhaps clarify the „majority vote“ (I take this to mean you pick the most-occurring label from training points routed to this leaf?). Also, what is „otherwise“ here – are the classifiers at leaves not defined to be labels...? i) p. 4, last par.: „...misclassification loss for a subset $\mathcal{C}_i$ ...“ – should it not be $\mathcal{S}_i$ here? Also, „...optimizing (2) where the misclassification error is summed over the whole training set is equivalent to optimizing it over the subset of training points $\mathcal{S}_i that reach node $i$.“ – and then summing the subset-errors over all $i$ ? (Else, I don't see this.) Please elaborate/be a bit clearer here. j) p. 5, 1st par.: So, is $\mathcal{C}_i$ a subset of $\mathcal{S}_i$ ? Perhaps include this relation at first occurrence of $\mathcal{C}_i$ to make it clear. k) p. 5, regarding „Sparse oblique trees“ – What about working with L0 directly rather than resorting to L1 as a sparsity proxy? (Naturally, this yields NP-hard problems again, but there are efficient iterative schemes that may work well in practice.) I don't expect the authors to work out the whole thing based on exact (L0-) sparsity measures, but I would like to see a brief discussion of this aspect at this point. l) In the suppl. mat., Fig. 2: The legend is sticking out – perhaps it could just be moved into the „empty“ part of Fig. 2 at the bottom right? Finally, there are several typos, so please proof-read again and fix what you find (out of space to list all I found, but e.g.: "vs"/"vs.","eq."/"Eq.","fig./Fig.","inputed").

Reviewer 4



In this paper, the authors propose a new algorithm for optimizing tree. Given an input tree, the proposed algorithm produces a new tee with the same structure but new parameter values. Furthermore, it can learn sparse oblique trees, having a structure that is a subset of the original tree and few nonzero parameters Major issues: 1. The algorithm arms to optimize eq. (1) but assuming a given, fixed tree structure. So, its performance highly depends on the initial tree structure, and it is essentially a postprocessing algorithm. Since there exist techniques for pruning a given tree, it is unclear to me why the proposed algorithm is necessary. 2. The algorithm is heuristic, and lacks theoretical guarantees. Theorem 3.1 relies on a heavy assumption. Theorem 3.2 is about the classification error, instead of generalization error. Furthermore, the convergence behavior of TAO is unclear. 3. In the experiments, the authors did not compare with other methods. I am ok with acceptance, provided the authors could address the following concerns. 1. Since the algorithm is a postprocessing method. The authors should examine the impact of the initial tree structure on the performance. I notice that in the rebuttal the authors have provided more experiments. 2. I agree that it could be too difficult to provide theoretical guarantees of the proposed method. But the authors should try some methods, such as regularization, to avoid overfitting. In the current version, the issue of overfitting is left to cross validation. 3. For the binary misclassification loss in (4), does the selection of surrogate loss important?